Microbiology **Spectrum**
# Cysteine-Dependent Conformational Heterogeneity of *Shigella flexneri* Autotransporter IcsA and Implications of Its Function

Jilong Qin,[a,b] Yaoqin Hong,[a] Renato Morona,[b] Makrina Totsika[a]

aCentre for Immunology and Infection Control, School of Biomedical Sciences, Queensland University of Technology, Brisbane, Queensland, Australia
bSchool of Biological Sciences, University of Adelaide, Adelaide, South Australia, Australia

**ABSTRACT** *Shigella* IcsA is a versatile surface virulence factor required for early and late pathogenesis stages extracellularly and intracellularly. Despite IcsA serving as a model Type V secretion system (T5SS) autotransporter to study host-pathogen interactions, its detailed molecular architecture is poorly understood. Recently, IcsA was found to switch to a different conformation for its adhesin activity upon sensing the host stimuli by *Shigella* Type III secretion system (T3SS). Here, we reported that the single cysteine residue (C130) near the N terminus of the IcsA passenger had a role in IcsA adhesin activity. We also showed that the IcsA passenger (IcsAp) existed in multiple conformations, and the conformation populations were influenced by a central pair of cysteine residues (C375 and C379), which was not previously reported for any Type V autotransporter passengers. Disruption of either or both central cysteine residues altered the exposure of IcsA epitopes to polyclonal anti-IcsA antibodies previously shown to block *Shigella* adherence, yet without loss of IcsA intracellular functions in actin-based motility (ABM). Anti-IcsA antibody reactivity was restored when the IcsA-paired cysteine substitution mutants were expressed in an Δ*ipaD* background with a constitutively active T3SS, highlighting an interplay between T3SS and T5SS. The work here uncovered a novel molecular switch empowered by a centrally localized, short-spaced cysteine pair in the Type V autotransporter IcsA that ensured conformational heterogeneity to aid IcsA evasion of host immunity.

**IMPORTANCE** *Shigella* species are the leading cause of diarrheal-related death globally by causing bacillary dysentery. The surface virulence factor IcsA, which is essential for *Shigella* pathogenesis, is a unique multifunctional autotransporter that is responsible for cell adhesion, and actin-based motility, yet detailed mechanistic understanding is lacking. Here, we showed that the three cysteine residues in IcsA contributed to the protein's distinct functions. The N-terminal cysteine residue within the IcsA passenger domain played a role in adhesin function, while a centrally localized cysteine pair provided conformational heterogeneity that resulted in IcsA molecules with different reactivity to adhesion-blocking anti-IcsA antibodies. In synergy with the Type III secretion system, this molecular switch preserved biological function in distinct IcsA conformations for cell adhesion, actin-based motility, and autophagy escape, providing a potential strategy by which *Shigella* evades host immunity and targets this essential virulence factor.

**KEYWORDS** *Shigella flexneri*, IcsA, conformation heterogeneity, adhesin

Address correspondence to Jilong Qin, Jilong.qin@qut.edu.au, or Makrina Totsika, Makrina.totsika@qut.edu.au.

The authors declare no conflict of interest.

**S**higella is a Gram-negative bacterial pathogen that is estimated to cause 80 to 165 million cases of shigellosis, with over 600,000 deaths worldwide annually (1). One of the essential *Shigella* virulence determinants (2) is the unipolar distributed surface protein IcsA (formerly VirG) (3). In the human gut lumen, *Shigella* senses the bile salt deoxycholate (DOC) (4) via the Type III secretion system (T3SS) needle residing protein IpaD (5) to regulate IcsA's adhesin activity required for pathogenesis (6). Inside the

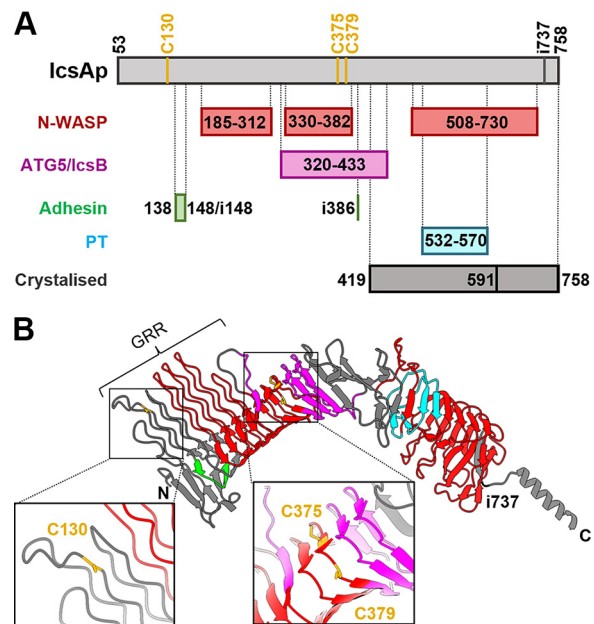

**FIG 1** The autotransporter protein IcsA passenger domain. (A) Schematic representation of defined subregions on IcsA passenger affecting its known biological functions with all three cysteine positions shown. PT, polar targeting. (B) IcsA passenger tertiary structure predicted by AlphaFold (AF-Q7BCK4-F1) with defined functional subregions marked in color as in (A) and cysteine residue locations is shown in a close-up view. GRR, glycine-rich repeats (aa 117 to 307).

host cells, *Shigella* uses IcsA to perform actin-based motility (ABM) for inter- and intra-cellular spreading by interacting with the host neural-Wiskott Aldrich syndrome protein (N-WASP) (7). In addition, IcsA was also found to be a target of host autophagy recognized by ATG5, a critical host protein for the initiation of autophagosome formation (8). However, *Shigella* efficiently escapes host autophagy by masking the IcsA region targeted by ATG5 with the *Shigella* T3SS effector protein IcsB (8) (Fig. 1A).

IcsA is encoded by the *Shigella* virulence plasmid (2) and characterized as a type Va autotransporter (9) with typical protein domain architecture, including an N-terminal signal peptide (aa 1 to 52), an ∼75 kDa central passenger domain (aa 53 to 758), and an anti-paralleled beta-barrel (aa 759 to 1102) that is embedded in the outer membrane (OM) for the translocation of the IcsA passenger (IcsAp). Approximately 20% of IcsAp is cleaved off by the OM protease IcsP after translocation, with the remaining having a unipolar localization on the bacterial surface (10). Protein subregions important for IcsA adhesin activity (6, 11), ABM function (12, 13), initiation of host autophagy (8), and bacterial unipolar targeting (14) were contained within the IcsAp domain (Fig. 1A). However, the structure of IcsAp was only available for its C-terminal portions (591 to 758 and 419 to 758) (15, 16). Thus, a detailed mechanistic understanding of how multiple functions were performed by this protein remains limited (Fig. 1A).

The translocation of the passenger domain in type Va autotransporters is generally assumed to occur through the hairpin model, where passenger translocation proceeds from the carboxyl terminus to the amino terminus through hairpin formation between a static strand sequestered in the lumen of the barrel and a sliding strand moving through the pore (17). This posits that the passenger remains unfolded in the periplasm, which has been supported by evidence whereby a large polypeptide loop formed by an intra-molecular disulfide bond before translocation inhibited the translocation of autotransporter passenger domain across the OM (18–21). This also agrees with the observation that most autotransporters have a low cysteine content in their passenger domain (22). However, nonautotransporter proteins containing a cysteine disulfide loop were shown to be successfully translocated when fused to autotransporter barrel domains (23, 24). A

few autotransporters have a single cysteine pair in their passenger domain, with the most common spacing being 11 residues and proximal to the C-terminal region of the passenger (25). Substitutions of such paired cysteine residues in the passenger domain of *Helicobacter pylori* VacA and *Serratia marcescens* Ssp-1 resulted in decreased protein production (25, 26). However, the role of cysteine residues in the passenger domain of type V autotransporters is poorly understood.

IcsA is a unique autotransporter and is distinctive to other cysteine-containing auto-transporters in that its paired cysteines are short-spaced (4 aa between C375 and C379) and located at the central region of the passenger, as well as the presence of an additional unpaired cysteine residue (C130) at the N-terminal region of IcsAp (Fig. 1A) (22). IcsA was reported previously to form an intramolecular disulfide bond in the periplasm before OM translocation in a DsbB-dependent manner (27). However, the exact cysteine residues participating in the intramolecular disulfide bond were not defined. In the AlphaFold predicted IcsA structure (Fig. 1B) (28), the unpaired C130 was located at the glycine-rich repeats (GRR) region close to the previously identified adhesin region (11) (Fig. 1A). The central paired cysteine residues (C375 and C379) are located at the beta-helical backbone near the ~90° kink region (16), which overlaps with both the N-WASP interacting region II (IRII) and the ATG5/IcsB binding region and are close to the other potential adhesin site (i386) identified previously (6, 8, 12) (Fig. 1A). However, no intra-molecular disulfide bond was represented in the predicted structure and whether these cysteines were important for IcsA biogenesis and/or biological functions remains unknown. Here, we investigated the role of the three cysteines in IcsAp biogenesis, conformation, and function. We showed that the unpaired cysteine residue C130 affected IcsA's adhesin activity and that the paired cysteine residues formed an intramolecular disulfide bond that impacted IcsA conformational heterogeneity.

## RESULTS

**Cysteine substitution IcsA passenger mutants had altered conformations.** We investigated the role of cysteine residues (C130, C375, and C379) within IcsAp in IcsA biogenesis and biological functions. We substituted each cysteine residue individually (C130S, C375S, and C379S) or in a pair (C375S/C379S; double mutant, DM) to serine to minimize potential structural perturbations. All substitutions did not affect IcsA expression levels (Fig. 2A). Interestingly, the substitution of the paired cysteine residues (C375 and C379) either individually or in combination, but not of unpaired cysteine residue (C130), abolished IcsA detection on the cell surface by indirect surface immunofluores-cent labeling with anti-IcsA polyclonal antibodies (Fig. 2B), which was shown to neu-tralize *Shigella* adherence to host cells (11). We reasoned that the loss of detection of the paired cysteine substitution mutants could be due to either a translocation defect or altered accessibility of epitopes recognized by the anti-IcsA antibodies used. To dis-tinguish between the two hypotheses, we performed proteinase surface shaving of intact *S. flexneri* bacteria expressing mutant IcsA proteins. We found that all IcsA cyste-ine substitution mutants remained sensitive to extracellular proteinase K digestion sim-ilar to the wild-type (WT) (Fig. 2C), suggesting that the passenger domain of IcsA was successfully translocated to the surface. Indeed, when we precipitated the secreted extracellular protein from culture supernatants, we found all IcsA cysteine substitution mutants were able to secrete cleaved IcsAp (Fig. 2D), confirming that passenger trans-location was not affected. Thus, the loss of surface detection of the IcsA-paired cysteine substitution mutants by anti-IcsA antibodies was most likely due to the altered accessi-bility of epitopes to anti-IcsA antibodies. To investigate this, and examine possible po-lar targeting defects, we engineered a FLAG × 3 tag at the C terminus of IcsAp (i737, Fig. 1) and confirmed that there was no impact on IcsA production (Fig. 2E). We then performed surface immunofluorescent labeling with both anti-IcsA and anti-FLAG anti-bodies. The anti-FLAG antibody, but not anti-IcsA antibodies, successfully labeled all IcsA cysteine substitution mutants and confirmed surface localization of IcsAp from the paired cysteine mutants (Fig. 2F). Image analysis confirmed that none of the cysteine

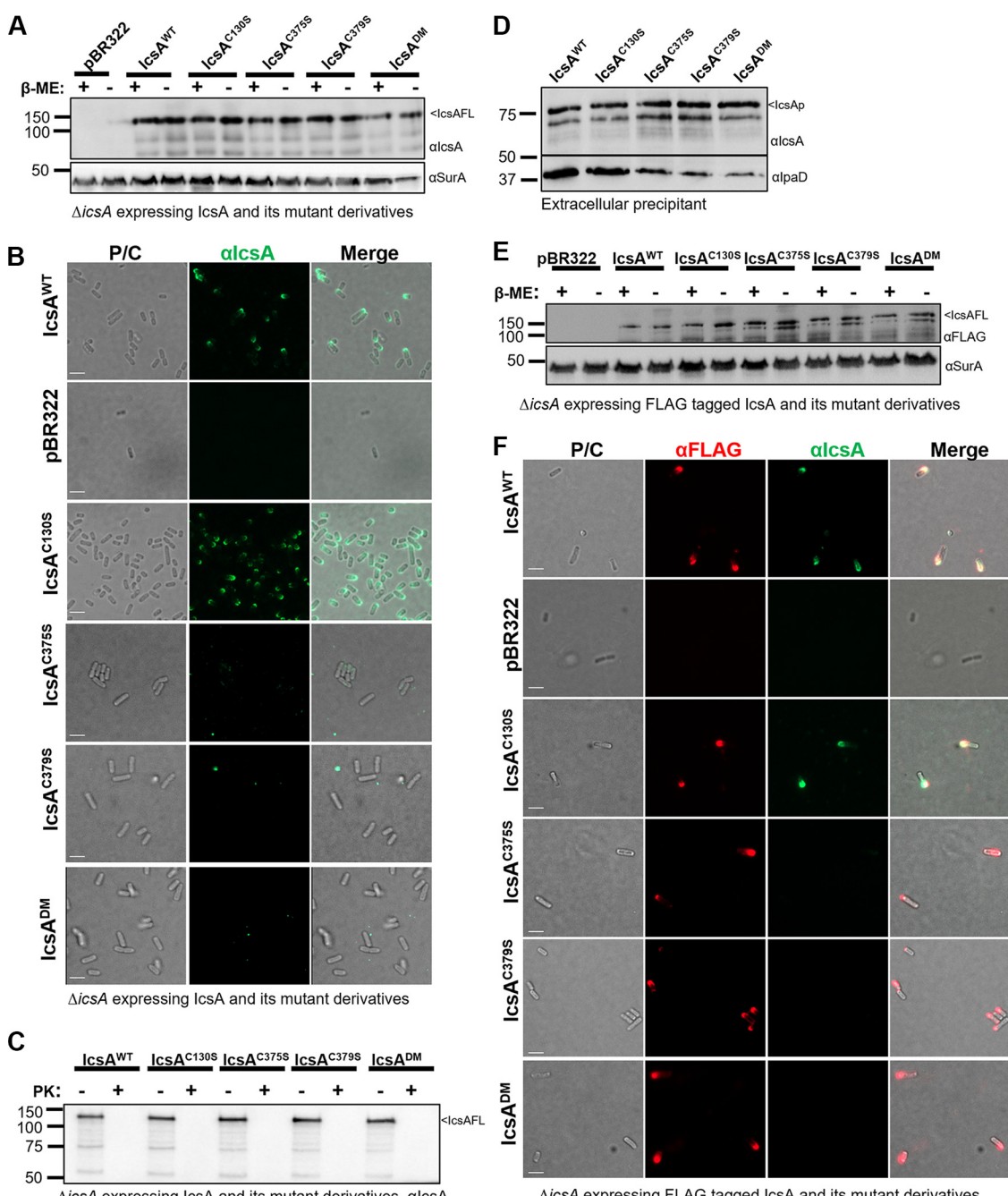

**FIG 2** Cysteine substitutions in IcsAp did not affect surface display but influenced protein conformation. (A) Western immunoblotting of bacterial cell lysates expressing IcsA^WT and cysteine mutant detected with anti-IcsA and anti-SurA antibodies. IcsAFL, IcsA full-length. (B) Indirect bacterial surface immunofluorescent labeling of *S. flexneri* expressing IcsA^WT and cysteine substitution mutants with anti-IcsA antibodies. (C) Western immunoblotting of IcsA with lysates of bacterial cells with and without proteinase K surface shaving. (D) Western immunoblotting of precipitated extracellular cleaved IcsAp with anti-IcsA and anti-IpaD antibodies. (E) Western immunoblotting of lysates of bacterial cells expressed FLAG-tagged IcsA^WT and cysteine substitution mutants with anti-FLAG and anti-SurA antibodies. (F) Indirect bacterial surface immunofluorescent labeling of FLAG-tagged IcsA passenger with anti-IcsA antibodies and anti-FLAG antibody. Scale bars are shown as 2 $\mu$m in (B and F).

substitution mutants had defects in polar targeting (Fig. 2F). Taken together, these data indicated that paired cysteine residues in IcsAp impacted IcsA conformation but not translocation, surface localization, or extracellular cleavage.

**Paired Cys substitution in IcsA altered the conformation of its N terminus and the conformational difference was maintained after secretion.** To characterize the IcsAp conformational differences due to cysteine substitution mutations on the

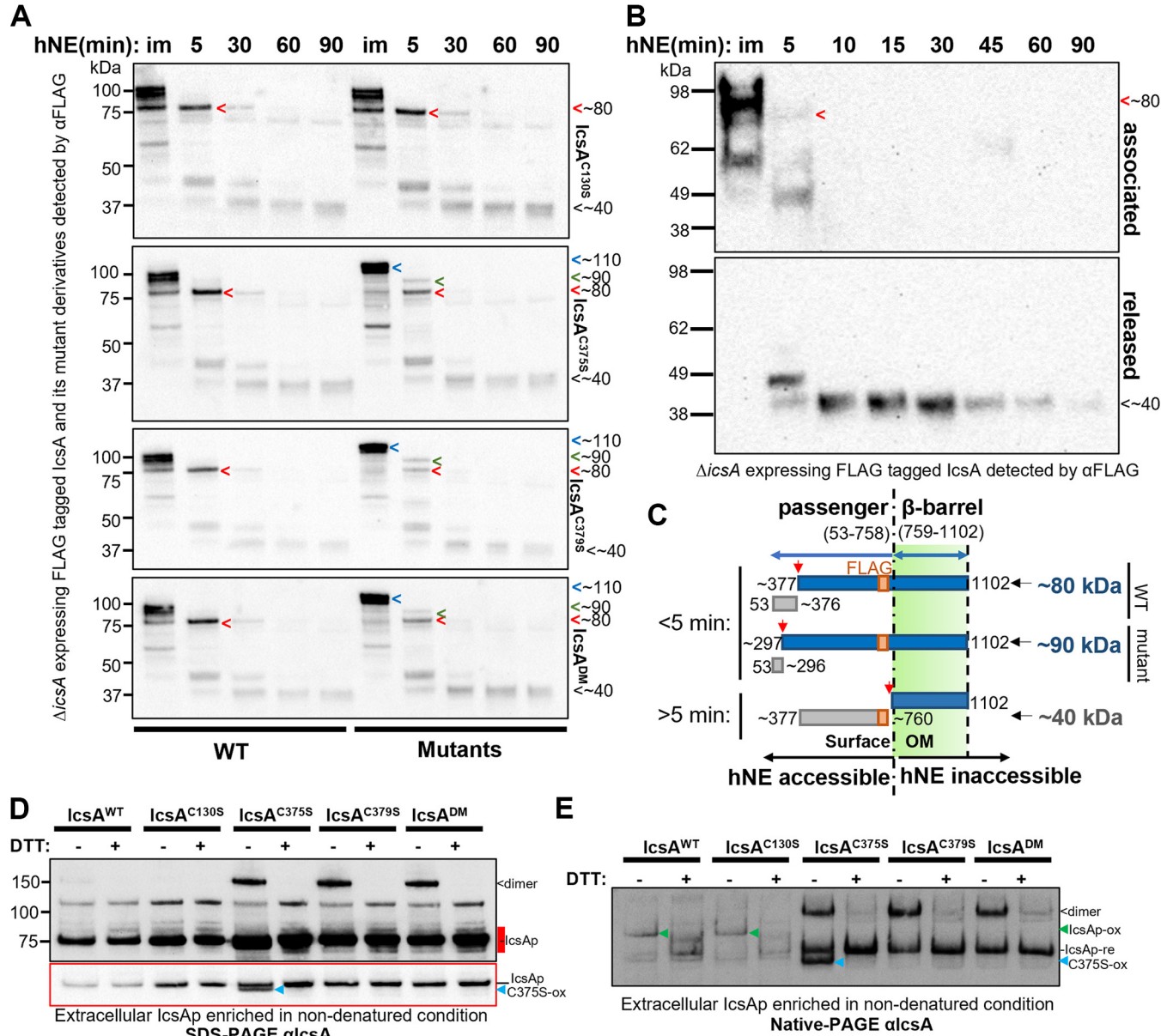

**FIG 3** The IcsA N terminus had the altered conformation and was maintained in the cleaved IcsAp. (A) Western immunoblotting with bacterial cells expressing IcsA and its cysteine substitution mutants treated with hNE. Samples were taken at different time points as indicated above the blot. im, immediate recovery of the digestion sample upon the addition of hNE. Samples from bacteria expressing IcsA[WT] were electrophoresed in parallel (left) with mutants (right) and blotted together to ensure equal exposure for direct comparison. (B) Western immunoblotting of fractionated hNE digestion samples from the bacteria expressing IcsA[WT]. (C) Model of the digestion differences due to altered conformation in IcsA mutant with estimated cutting sites shown. (D) Western immunoblotting of IcsAp separated in semidenatured conditions. Extracellular IcsA was precipitated in ammonium sulfate and was solubilized in SDS-PAGE buffer in the presence or absence of DTT and was subjected to SDS-PAGE and immunoblotting without heating. The underexposure image in the red border correlates to the region in the long exposure image marked red. re, reduced; ox, oxidized. <, IcsAp dimer; blue arrow, oxidized IcsA[C375S], C375S-ox. (E) Culture supernatant IcsAp protein recovered as above was analyzed by Native-PAGE and detected with anti-IcsA antibodies. Green arrow, oxidized IcsAp, IcsAp-ox.

bacterial surface, we performed limited proteolysis with human neutrophil elastase (hNE) to surface shave IcsAp in a pulse-chase manner. Under immediate exposure of hNE, the IcsA[WT] and IcsA[C130S] were digested in an ~100 kDa fragment followed by an ~80 kDa fragment (Fig. 3A, red arrow) at 5 min, and an ~40 kDa fragment for the rest of 85 min digestion duration (Fig. 3A, black arrow). These observations suggested that the hNE accessibility was not detectably altered between IcsA[WT] and IcsA[C130S]. In contrast, all paired cysteine substitution mutants were more resistant immediately after exposure to hNE, producing a fragment running higher than that of IcsA[WT] (~110 kDa)

(Fig. 3A, blue arrow) followed by an ~90 kDa fragment at 5 min (Fig. 3A, green arrow). The ~40 kDa band (Fig. 3A, black arrow) appeared after 5 min digestion and was consistently detected for all IcsA mutants and IcsA$^{WT}$.

To further distinguish the cell association states of these FLAG-tagged IcsA fragments, we cell fractionated the digestion reaction for each time point, and Western blotting revealed that the ~40 kDa FLAG-tagged fragment was cleaved off the bacterial cell surface (Fig. 3B). Hence, it can be deduced that the FLAG-tagged IcsA fragment at ~40 kDa was devoid of the beta-barrel domain (inaccessible to hNE) because it was not cell-associated. HNe was reported previously to preferably cleave valine, alanine, and isoleucine (29). Therefore, it could be estimated that the ~40 kDa fragment was derived from aa 377 to 760 (FLAG engineered at i737) (Fig. 3C), suggesting that the accessibility to hNE in this region was unchanged for all cysteine substitution mutants. In contrast, the larger FLAG-tagged IcsA ~80 kDa fragment remains cell-associated (Fig. 3B, red arrow) and, therefore, was uncleaved from the beta-barrel. Hence, it could be estimated to be derived from aa 377 to 1102 (Fig. 3C). Similarly, the ~90 kDa FLAG-tagged fragment detected from the paired cysteine substitution mutant IcsA can be estimated to be derived from aa 297 to 1102 (Fig. 3C). These data indicated that the altered hNE accessibility of IcsAp for all paired cysteine substitution mutants was at the N-terminal region of the protein, between aa 53 and 376, just before the C375-C379 pair. This prediction also agreed with the differential surface immunostaining observed with the anti-IcsA antibodies previously (Fig. 2B and F), which were confirmed to only react with IcsAp (30).

IcsAp can be cleaved off the bacterial surface by IcsP and be secreted into the extracellular medium (10). To rule out the possibility that the observed changes for the paired cysteine substitution mutants were due to differential LPS masking, we compared all secreted IcsA cysteine substitution mutant passengers to that of the IcsA$^{WT}$ in semidenatured conditions in the presence or absence of the disulfide bond reducing agent DTT (Fig. 3D). As expected, following recovery from the culture supernatant, all cleaved IcsAp were detected at the molecular size of ~75 kDa (Fig. 3D, IcsAp). Interestingly, a DTT-sensitive IcsAp population with a molecular size that correlates to an IcsA dimer (150 kDa, Fig. 3D, dimer) was detected with IcsA$^{WT}$ and all paired cysteine substitution mutants but not with the IcsAp$^{C130S}$, suggesting that C130 was involved in an intermolecular disulfide bond formed between two IcsAp molecules under our experimental conditions. We also detected a faster migrating and DTT-sensitive IcsAp band only with the IcsA$^{C375S}$ mutant (Fig. 3D, C375S-ox). Presumably, this was due to the oxidation of the two remaining cysteines C130 and C379, leading to a slightly more compact molecule under semidenatured conditions. This also suggested that C130 was located close to C379 in the folded IcsAp structure.

Next, to capture differences in protein conformation mediated by the paired cysteines that could not be potentially resolved under semidenatured conditions due to the small size of the disulfide loop formed by C375-C379, we compared all cysteine substitution mutants in nondenatured gel conditions (Fig. 3E). Under these conditions, migration speed potentially changed for proteins in different conformations due to changes in molecular shape, oligomeric state, and exposure of their intrinsic surface charge. Under nondenaturing conditions, we still detected the potential dimeric form of IcsAp, which was predominantly enriched in the paired cysteine substitution mutants (Fig. 3E, dimer) and the oxidized IcsAp$^{C375S}$ (Fig. 3E, C375S-ox). In addition, we detected a DTT-resistant IcsAp for IcsAp$^{WT}$ and all cysteine substitution mutants (Fig. 3E, IcsAp-re). In contrast, IcsAp$^{WT}$ and the IcsAp$^{C130S}$ were instead detected with an additional DTT-sensitive IcsAp form migrating at a different speed (Fig. 3E, IcsAp-ox) relative to those detected with paired cysteine mutant IcsAp. The populations (Fig. 3E, IcsAp-ox) likely observed for IcsAp$^{WT}$ and IcsAp$^{C130S}$ were due to intramolecular oxidation between C375 and C379 because this population was absent in the paired cysteine substitution mutants (Fig. 3E, IcsAp-re). Together, these data suggested that IcsA

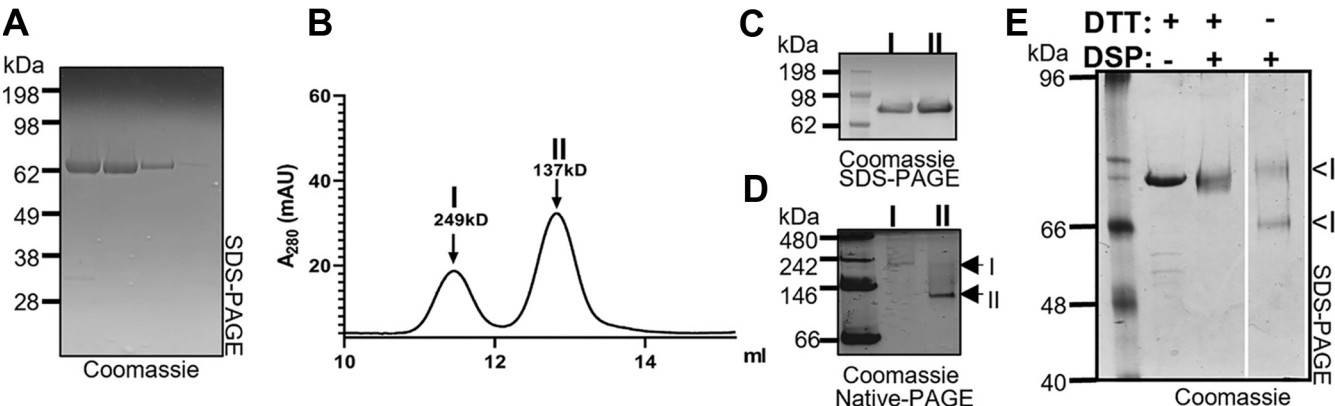

**FIG 4** Purified IcsA existed in two different conformations. (A) Affinity purified IcsAp resolved by SDS-PAGE and stained with Coomassie blue. (B) SEC separation of IcsAp into populations I and II. (C) SDS-PAGE of IcsAp populations I and II isolated by SEC. (D) Native-PAGE of IcsAp populations I and II isolated by SEC. (E) SDS-PAGE of IcsAp cross-linked with 0.1 mM DSP in the presence or absence of 10 mM DTT. The image was from the same gel with the relevant area shown.

conformation was maintained through a disulfide bond formed between C375 and C379 and that the IcsAp conformation was maintained even after IcsP cleavage.

**Purified IcsA existed in two conformations.** To show that the conformations of IcsAp were independent of binding to other secreted *S. flexneri* effector proteins, we adapted a similar expression and purification strategy to that used previously (31) to isolate native N-terminally FLAG-tagged IcsAp (Fig. 4A) from the culture supernatant. Same as reported previously, we separated IcsAp into two fractions through size exclusion chromatography (SEC) (Fig. 4B and C) (31). The second fraction (migrating as 137 kDa determined with native molecular weight markers with the globular shape) could be assigned to a monomer in agreement with a previous study (31), where multiangle light scattering (SEC-MALS) was used. These two IcsAp species could also be separated using Native-PAGE (Fig. 4D), where we found that the size at which each species migrated correlated with the results obtained from our SEC experiment (Fig. 4B). We were aware that the protein molecular weight determination with the globular molecular weight markers was irrelevant because the shape of IcsAp was predicted to be elongated. Hence, we were only using them as a relative mass reference to correlate our SEC and Native PAGE results.

To prove that purified IcsAp has two forms adopting different protein conformations, we attempted to fix any close intramolecular contacts by incubating purified IcsAp protein with the DTT-reducible cross-linker DSP (Fig. 4A) and comparing the migration profile under denaturing gel conditions. Upon cross-linking with DSP, we resolved an IcsAp population with a faster migration (Fig. 4e, II), suggesting a compact conformation and a closer intramolecular contact than the alternative IcsAp conformation (Fig. 4e, I). Together, these data suggested that the IcsAp existed in multiple conformations with a compact form that allowed closer intramolecular contacts and an extended form with fewer intramolecular contacts.

**Effects of IcsA cysteine substitution mutations in *Shigella* host cell adherence and infection.** IcsA was well known as a multifunctional virulence factor with defined regions interacting with different host molecules (Fig. 1A), facilitating *Shigella* pathogenesis at different stages. Here, we established that the cysteine substitutions in the IcsA passenger domain altered IcsA oligomeric states and conformations and even altered the accessibility of polyclonal anti-IcsA antibodies. We then investigated whether these altered conformations due to cysteine substitution would also alter the interactions of IcsA with host molecules, thereby having functional implications. IcsA was reported previously to act as an adhesin interacting with host cell surfaces when T3SS was activated (6). We, therefore, expressed all IcsA cysteine substitution mutants in an *S. flexneri* Δ*ipaD* strain background, where the loss of IpaD results in hyper-adhesion. This was similar to IpaD binding to the host environmental stimulus deoxycholate

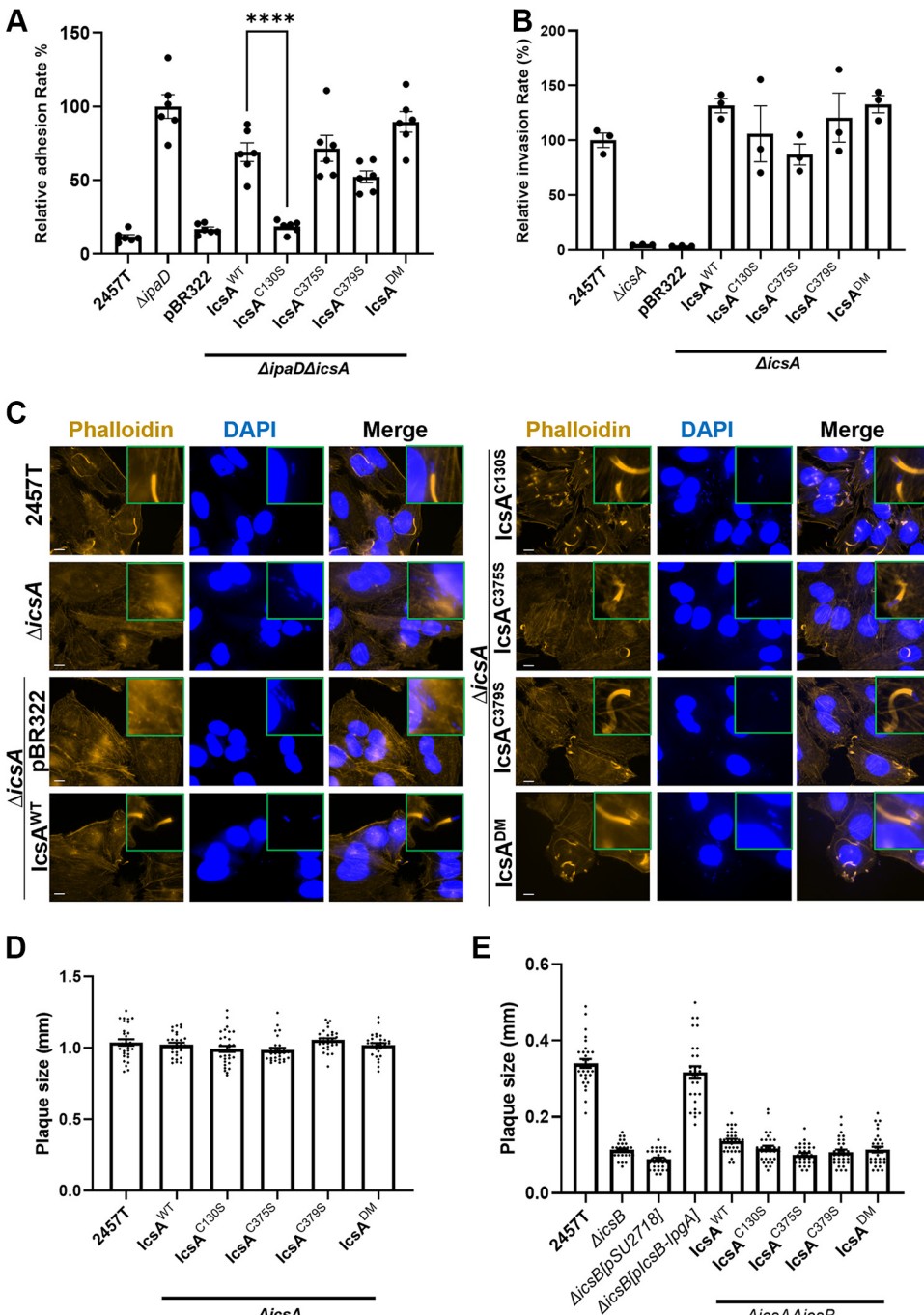

**FIG 5** Effect of cysteine substitution mutation on IcsA function. (A) Cell adhesion assay of indicated *S. flexneri* strains with HeLa cells. Mean values from 6 independent assays with standard error mean (SEM) are shown. Statistical significance was calculated using one-way analysis of variance (ANOVA) followed by Dunnett's multiple-comparison test against Δ*ipaD*Δ*icsA* pIcsA$^{WT}$. ****, $P < 0.0001$. (B) HeLa cell invasion assay of indicated *S. flexneri* strains treated with 2.5 mM DOC. Data represent 3 independent assays. (C) Fluorescent labeling of host cell F-actin by phalloidin and host and bacterial nuclei by DAPI of HeLa cells infected with indicated *S. flexneri* strains. Scale bar is 10 μm. (D and E) Plaque sizes formed by indicated *S. flexneri* strains in MDCK-2 monolayers. Data were from two independent assays with ~30 plaque sizes plotted for each strain.

(DOC). The strain producing IcsA$^{C130S}$, but not strains producing paired cysteine substitution IcsA mutants (IcsA$^{C375S}$, IcsA$^{C379S}$, and IcsA$^{DM}$), lost hyper-adhesion to host cells in comparison to controls with IcsA$^{WT}$ (Fig. 5A). However, none of the strains producing IcsA mutants had a defect in cell invasion (Fig. 5B). In addition, although the paired

cysteine residues were in the N-WASP protein interaction region II (Fig. 1A), which was previously reported to be important for ABM function (12), we observed no differences in intracellular F-actin polymerization (Fig. 5C) for the strains with IcsA cysteine substitution mutants compared to the WT. This was also supported by the plaque formation assay which showed no defects in intercellular spreading ability owing to the normal ABM function for strains with IcsA mutants (Fig. 5D and Fig. S1A in Supplemental File 1). The region where C375 and C379 were located (Fig. 1A) was also reported to be a target of host autophagy through the recognition and binding of the host autophagosome component ATG5, yet intracellular *Shigella* can escape this host defense mechanism by masking the same region on IcsA by binding to IcsB (8). Indeed, *S. flexneri* lacking IcsB produced smaller plaques, and this could be restored by *trans* complementation of IcsB and its fusion chaperone complexes (Fig. 5E and Fig. S1B in Supplemental File 1) as reported before (32). We then examined if the altered conformation in IcsAp seen with the paired cysteine mutants would be naturally resistant to host autophagy without IcsB masking due to any potentially altered accessibility to host ATG5. We did not detect any restored plaque sizes for any strains with IcsA cysteine substitution mutants in the absence of IcsB masking (Fig. 5E and Fig. S1B in Supplemental File 1). Taken together, our data suggested that the C130 residue of IcsA was involved in the adhesin function, consistent with its location close to the previously defined adhesion region (11), while the altered conformations due to the paired cysteine substitutions were tolerant to ABM function and escape from autophagy under our experimental conditions.

**The ΔipaD mutation restored anti-IcsA accessibility by further altering IcsA conformation.** The IcsA paired cysteine substitution mutants (IcsA$^{C375S}$, IcsA$^{C379S}$, and IcsA$^{DM}$) exhibited dramatic conformational changes that masked epitopes reactive with the anti-IcsA antibodies used, yet the proteins were still able to polymerize F-actin and be the target of host autophagy when exposed intracellularly. It was known that the *Shigella* type III secretion system (T3SS) can be activated through the binding of IpaD to DOC, which in turn alters IcsA conformations (6). We also confirmed the IcsA conformational changes when expressed in our *S. flexneri* ΔipaDΔicsA background via limited proteolysis (33). Potentially such conformational changes may rescue the altered accessibility to anti-IcsA antibodies in the paired cysteine substitution mutants. Indeed, when expressed in an *S. flexneri* ΔipaDΔicsA background with constitutively active T3SS, the surface detection of IcsA$^{C375S}$, IcsA$^{C379S}$, and IcsA$^{DM}$ by anti-IcsA antibodies was restored (Fig. 6A). However, the pattern of hNE digested IcsA fragments of *S. flexneri* ΔipaDΔicsA expressing IcsA$^{C375S}$, IcsA$^{C379S}$ and IcsA$^{DM}$ remain to be different to that of IcsA$^{WT}$ and IcsA$^{C130S}$ (Fig. S2 in Supplemental File 1). We then investigated the conformation of secreted IcsAp of all the IcsA mutants in both semidenatured (Fig. 6B) and nondenatured conditions (Fig. 6C). We consistently detected the IcsAp population with the molecular mass correlated with a dimerized IcsA complex (in a C130 dependent manner) (Fig. 6B, dimer), as well as the oxidized IcsAp$^{C375S}$ population only with the strain expressing IcsA$^{C375S}$ (Fig. 6B, C375S-ox). However, the amount of the secreted IcsAp dimer population (Fig. 6B and C, dimer) for the paired cysteine substitution IcsA mutants was decreased when expressed in the hyper-adhesive strain *S. flexneri* ΔicsAΔipaD in comparison to *S. flexneri* ΔicsA (Fig. 6C, dimer, and Table 1). In contrast, the level of the other conformation population (Fig. 6C, left, IcsAp-re) for the paired cysteine substitution IcsA mutants was increased and was resistant to DTT treatment (Fig. 6C, right, IcsAp-re). A similar conformation population increase in *S. flexneri* ΔicsAΔipaD was also observed for IcsAp$^{WT}$ and IcsAp$^{C130S}$, albeit in an oxidized form (Fig. 6C, left, IcsAp-ox). The oxidized form detected in IcsAp$^{WT}$ and IcsAp$^{C130S}$ was potentially mediated by a disulfide bond between C375 and C379 (Fig. 6C, left, IcsAp-ox) because they migrate the same as the paired cysteine substitution mutant IcsAp upon DTT reduction (Fig. 6C, right, IcsAp-re). Collectively, IcsAp expressed from *S. flexneri* ΔicsAΔipaD favors a conformation with fewer intermolecular interactions with a decrease in the C130-mediated IcsA dimer population. These data suggested that the

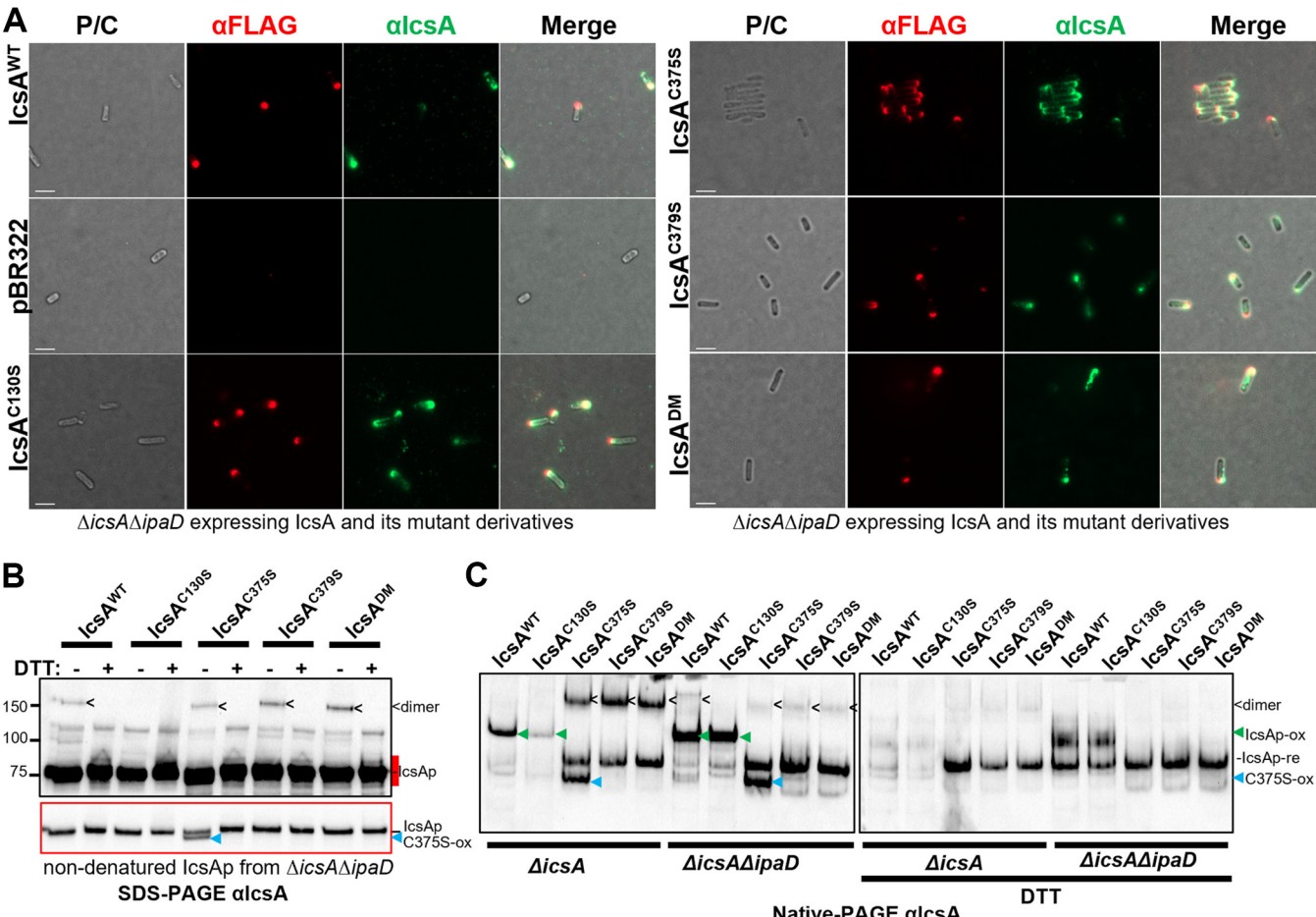

**FIG 6** Effect of ΔipaD on IcsA conformation. (A) Indirect bacterial surface immunofluorescent labeling of FLAG-tagged IcsA passenger with anti-IcsA antibodies and anti-FLAG antibody. (B) Western immunoblotting of IcsAp recovered from the culture supernatant of indicated *S. flexneri* strains expressing IcsA and IcsA mutants in nondenatured conditions. Samples were solubilized in SDS-PAGE buffer with or without DTT and subjected to SDS-PAGE and Western immunoblotting without heating. The underexposure image in the red border correlates to the region in the long exposure image marked red. re: reduced, ox: oxidized, <: IcsAp dimer; blue arrow: oxidized IcsA$^{C375S}$, C375S-ox. (C) IcsAp protein recovered as above and analyzed by Native-PAGE and detected with anti-IcsA antibodies. Green arrow: oxidized IcsAp, IcsAp-ox.

activation of T3SS altered the conformation landscape of IcsA to restore its epitope exposure to anti-IcsA antibodies used.

## DISCUSSION

IcsA is a multifunctional virulence factor being strictly required for different stages of *Shigella* pathogenesis. The configuration of bearing a central cysteine pair and an N-terminal unpaired cysteine residue makes IcsA a unique type Va autotransporter. We here studied the role of these three cysteine residues on IcsA's structure and biological functions (Table 1).

We discovered that the unpaired cysteine residue C130 was involved in IcsA's adhesin activity. C130 is located close to the previously identified region (138 to 148) affecting IcsA adhesin function (11). Deletion of either this entire region or insertion by a linker (5aa) at Q148 affected IcsA adhesin activity (6, 11). The unpaired cysteine residue C130 may be directly contributing to the host cell adherence because the disruption of the upstream folding at the C terminus would then affect the subsequent folding of this region during the later stage of OM translocation and may orient C130 differently. In support of this speculation, substitution Q148C also completely abolished IcsA's adhesin activity, possibly due to disulfide bond formation between C148 and C130 (11). A single cysteine residue has also been reported previously to be responsible for the activity of adhesin Ifp in *Yersinia pseudotuberculosis* (34).

**TABLE 1** Cysteine-dependent IcsA multiconformations and their functional implications[a]

| IcsA type | *S. flexneri* Δ*icsA* | | | | | *S. flexneri* Δ*icsA*Δ*ipaD* | | | | |
|---|---|---|---|---|---|---|---|---|---|---|
| | WT | C130S | C375S | C379S | DM | WT | C130S | C375S | C379S | DM |
| Anti-IcsA reactivity | + | + | − | − | − | + | + | + | + | + |
| ABM | + | + | + | + | + | nt | nt | nt | nt | nt |
| Adhesion | nt | nt | nt | nt | nt | + | − | + | + | + |
| Invasion | + | + | + | + | + | nt | nt | nt | nt | nt |
| AE with Δ*icsB* | − | − | − | − | − | nt | nt | nt | nt | nt |
| | | | | | | | | | | |
| IcsAp conformations | | | | | | | | | | |
| IcsAp-re | low | low | high | high | high | low | low | high | high | high |
| IcsAp-ox | high | high | nd | nd | nd | high | high | nd | nd | nd |
| Dimer | low | nd | high | high | high | low | nd | low | low | low |
| C375S-ox | low | low | high | nd | nd | low | low | high | nd | nd |

[a]The relative level of IcsAp conformation populations detected by Western immunoblotting was reported. AE, autophagy escape; DM, C375S/C379S double mutant; nt, not tested; nd, nondetectable.

We also detected an IcsA dimer population that was formed by a disulfide bond between C130 residues from two different protein molecules. It is unlikely that the intermolecular disulfide bond was formed in the periplasm. Instead, it is likely to be due to posttranslocation oxidation by the environment. IcsA has been reported previously to have self-association activity (35), and mutants were demonstrated to restore the IcsA N-WASP binding defect cooperatively (13), yet the exact region for the self-association remains unclear. Such self-association may orient the two C130 residues from two IcsA molecules to form a disulfide bond.

By substituting the paired cysteine residues with serine, either individually or together, we revealed that the centrally localized cysteine pair (C375/C379) form an intramolecular disulfide bond. IcsA has been reported previously to possess an intramolecular disulfide bond catalyzed by the periplasmic disulfide bond formation enzymes DsbB/DsbA before OM translocation (27). Here, we showed that the intramolecular disulfide bond in IcsAp was maintained after the OM translocation and its subsequent release into the extracellular milieu by IcsP. It has been shown previously with *Helicobacter pylori* VacA (25) and *S. marcescens* Ssp-1 (26) that disruption of intrinsic paired cysteine residues led to decreased protein production. However, we showed here that disruption of paired cysteine residues in IcsAp did not influence protein production. We reasoned that this was due to both the localization of and the spacing between the paired cysteine residues. Both VacA and Spp-1 have a pair of cysteine residues localized to the C terminus of the passenger domain (25), which potentially is required to stabilize the initial translocation hairpins for efficient translocation of the whole passenger. In contrast, the central short-distanced cysteine pairs in IcsA are likely to be tolerated during later translocation events where the C-terminal beta-helical stable core (15) has been completely translocated. This is also shown in other autotransporters with engineered short-spaced cysteine pairs, where small cysteine loops are tolerated during the OM translocation of the autotransporter passengers (18, 21). However, we discovered that these short-spaced cysteine pairs in the IcsAp instead influenced the autotransporter conformation when displayed on the cell surface and may act as a molecular constrain to influence the matured autotransporter conformations after translocation. This may be similar to those adjacent cysteine residues in nonautotransporters, which were shown previously as a redox switch to allosterically influence protein conformations and function (36, 37). By comparing the differential exposure of proteolysis sites to hNE for all IcsA cysteine substitution mutants, we refined the region with altered conformation in those paired cysteine mutants to be limited to aa 52 to 376, which was at the N-terminal side rather than the C-terminal side of C375/C379. This is probably due to the order of passenger translocation, where the C-terminal beta-helical core is first folded to guide the subsequent folding of N-terminal regions. Disruption of the disulfide bond formation between C375 and C379 is likely to alter the local folding and influences

the subsequent N-terminal protein conformations only. Our findings may also be a useful guide for the design of short-spaced cysteine pairs in autotransporter passengers to study their protein conformations, functions, and generalized folding mechanisms.

In addition, we showed in detail that one of the purified IcsAp conformation populations had a close intramolecular contact, which upon cross-linking, fixed it into a compact molecule in denatured conditions. This was distinct from the other conformational populations. We speculate that this intramolecular interaction may occur between the central region and the N-terminal region of the IcsAp through a potential bridging loop as proposed previously (31), although it was not predicted by the Alphafold (Fig. 1B) (28), which instead predicted IcsA is having no cysteine residues forming an intramolecular disulfide bond. The former is, however, supported by our experimental data. Specifically, upon disruption of C375, C130 likely forms a disulfide bond with C379, suggesting an intramolecular contact that oriented these two residues from N-terminal and central passenger closely in space. This result may also partially explain the previous report where a linker insertion at position i386 also affected IcsA adhesin activity (6) because they might have close contact with N-terminal regions.

IcsA holds great vaccine candidate potential because it was demonstrated previously that the level of antibodies to IcsA significantly reduces the risks of shigellosis (38). High levels of anti-IcsA antibodies were also found as one major maternal antibody transferred to the infant and may likely contribute to disease prevention during the first months of life (39). The correlated protection is likely to be through the prevention of early *Shigella* host cell attachment as was demonstrated previously (11) with adhesion-blocking activity by anti-IcsA antibodies and not through antibody-mediated complement killing (39). However, we showed that the binding of the adhesion-blocking polyclonal anti-IcsA antibodies to IcsAp could be abolished upon the disruption of the centrally localized cysteine pair (C375 and/or C379) in IcsAp, which was involved in intramolecular disulfide bond formation. This was due to the conformational heterogeneity of IcsA. Indeed, it has been reported previously that a monoclonal anti-IcsA antibody targeting the GRR region only stained 40% of the *Shigella* bacterial cell surfaces (40). We also showed that the IcsAp subregion responsible for its conformational heterogeneity was estimated to be aa 53 to 376, overlapping with the entire GRR region. More importantly, this MAb work suggested that the differences in IcsA conformations were probably segregated at the single-cell level. Indeed, using a different polyclonal anti-IcsA antiserum, IcsA was only stained for 50% on *S. flexneri* 5a (MT90) (41). In addition, disruption of DegP, a periplasmic chaperone, in *Shigella flexneri* reduced accessibility of different polyclonal anti-IcsA antibodies (produced independently to this work) by 40%, without impairing IcsA protein production (42). These data suggested that the conformational heterogeneity of IcsA with different reactivity to antibodies was not limited to our in-house made antiserum and the heterogeneity was controlled before OM translocation. Here, we showed that IcsA's conformation landscape was influenced by an intramolecular disulfide bond formed between the central cysteine pair, and disruption of the disulfide bond by substitution mutagenesis shifted the IcsA population into a conformation with no reactivity to the adhesion blocking antibodies (Table 1). However, in this conformation, we could not detect any defects in their cell infection function, including cell adhesion, invasion, and intracellular movement. The cellular function of IcsA upon paired cysteine disruption might be also restored due to the further conformational changes such as upon the activation of T3SS as shown with Δ*ipaD* here, in a similar way to the restoration of antibody detection. IcsA is an essential virulence factor required for *Shigella* pathogenesis at both early and late infection stages and is one of the major antigens targeted by host humoral immunity postinfection (39), yet it is highly conserved in amino acid sequence. Together, we propose a model that the conformational heterogeneity of IcsA found in this work empowered by a central pair of cysteine residues may be a strategy evolved by *Shigella* IcsA to evade host immunity by having multiple conformations with different reactivity to

antibodies elicited by the host, working closely with the T3SS to preserve its biological functions.

## MATERIALS AND METHODS

**Bacterial strains, plasmids, and mammalian cell line.** The bacterial strains and plasmids used in this work are listed in Table S1 in Supplemental File 1. Single colonies of *Escherichia coli* bacterial strains grown overnight on Lennox Broth (LB) (43) agar (1.5% wt/vol) plates or single red *S. flexneri* colonies grown on trypticase soy agar plates containing 0.03% (wt/vol) Congo red were picked and grown overnight in LB at 37°C for all experiments. Where appropriate, the medium was supplemented with ampicillin (Amp, 100 $\mu$g/mL), kanamycin (Kan, 50 $\mu$g/mL), streptomycin (Strep, 100 $\mu$g/mL), or chloramphenicol (Chl, 25 $\mu$g/mL). HeLa cells were routinely cultured and maintained with MEM (Gibco) supplemented with 5% (vol/vol) Fetal bovine serum (FBS), and Madin-Darby canine kidney-2 (MDCK-2) cells were cultured and maintained with MEM(Gibco) or DMEM (Gibco) supplemented with 5% (vol/vol) FBS (Gibco) at 37°C with 5% $CO_2$.

**Mutagenesis by allelic exchange and inverse PCR.** *S. flenxeri* $\Delta$*icsB* and $\Delta$*icsA*$\Delta$*icsB* mutants were generated by Lambda Red mutagenesis as described previously (44) with the oligonucleotides listed in Table S1 in Supplemental File 1.

**Plasmid construction.** For the IcsB expression construct, the coding sequence of IcsB and its chaperone protein IpgA along with their native promoter region was PCR amplified (Table S1 in Supplemental File 1) and restriction cloned into pSU2718 with XbaI and KpnI, resulting in pIcsB-IpgA.

For IcsA site-directed mutagenesis, pIcsA (45) was used as the template to amplify the entire plasmid by inverse PCR using oligonucleotides (Table S1 in Supplemental File 1) with cysteine to serine amino acid substitutions at the 5′ ends. Amplicons were then phosphorylated, ligated, and introduced into TOP10 for recovery. Plasmids were extracted from transformants, and the codon substitution was confirmed by sequencing. The in-frame addition of FLAG $\times$ 3 affinity tag after aa 737 (i737) was engineered using the same method as above with oligonucleotides (Table S1 in Supplemental File 1) targeting the insertion sites and in-frame fused with the coding sequence of FLAG $\times$ 3 at the 5′ ends. Fragments containing the amino acid substitutions were moved from pIcsA constructs into pIcsA$^{737::FLAG}$ by restriction cloning with XbaI and HindIII to generate FLAG-tagged IcsA substitution mutant constructs.

For extracellular IcsA passenger production, sequences of *icsA* and *icsP* were PCR amplified (Table S1 in Supplemental File 1) from *S. flexneri* 2457T chromosomal DNA, digested with NcoI/SalI and NdeI/KpnI, respectively, and then ligated into the MCS1 and MCS2 of the pCDFDuet-1 (Novagen) plasmid sequentially digested in the same way to generate pCDFDuet-1::*icsA-icsP*. A FLAG $\times$ 3 tag was in-frame inserted after aa 54 (i54) via restriction-free cloning (46) with oligonucleotides (Table S1 in Supplemental File 1) designed to assemble gene blocks containing the coding sequence FLAG $\times$ 3 tag flanked with sequence upstream and downstream of the i54 via PCR using each other as the templates. The gene block was then used as megaprimer pairs to PCR clone the FLAG $\times$ 3 tags at i54 using pCDFDuet-1::*icsA-icsP* as the template to generate the coexpression construct pCDFDuet-1::*icsA$^{54::FLAG}$-icsP* (pIcsA-IcsP).

**IcsA passenger purification and size exclusion chromatography.** For IcsAp (IcsA$^{54::FLAG}$) production, C43(DE3) coexpressing IcsP and FLAG-tagged IcsA were subcultured 1 in 20 into 4 L of LB from 18 h cultures and were grown to an optical density at 600 nm (OD$_{600}$) of 0.4 to 0.6 at 37°C. Expression of IcsA was then induced with 1 mM isopropyl $\beta$-D-1-thiogalactopyranoside (IPTG) and cultures were incubated at 30°C for 18 h. The culture supernatant was harvested via centrifugation (7,000 $\times$ *g*, 25°C, 15 min), and further cleaned by passing through an asymmetric polyethersulfone (aPES) membrane with 0.2 $\mu$m pore size (Rapid-flow, Thermo Scientific). Proteins in the filtered culture supernatant were then concentrated and exchanged into 50 mL of TBS buffer (50 mM Tris, 150 mM NaCl, pH 7.5) using a VivaFlow 200 filtration system (Sartorius) with 30,000 MWCO PES membranes. The concentrated protein preparation was then loaded onto a TBS preequilibrated polypropylene column (Thermo Scientific) prepacked with 2 mL of anti-FLAG G$_1$ resin (Genescript). The column was washed with TBS, and IcsA passenger was eluted with 10 mL of 100 mM glycine, pH 3.5. IcsA passenger was then concentrated to 500 $\mu$L using a Vivaspin 6 with 10 kDa MWCO (GE Healthcare). For size exclusion chromatography analysis, the protein was loaded onto a Superdex 200 Increase 10/300 column (GE Healthcare) with TBS buffer, and different protein elution fractions were pooled and concentrated again as above, which yielded ~0.8 mg from 4 L culture supernatant. Sizes were only referenced according to the Gel Filtration Makers kit (for protein molecular weight 29 kDa to 700 kDa, Sigma).

**Polyacrylamide gel electrophoresis and Western immunoblotting.** For the analysis of proteins under denatured conditions, bacterial whole-cell lysate was prepared by harvesting bacteria (~5 $\times$ 10$^8$ cells) grown to the mid-exponential phase (OD$_{600}$~0.4 to 0.8) via centrifugation (16,000 $\times$ *g*, 1 min). The pellet was resuspended into 100 $\mu$L SDS-PAGE sample buffer (47) and heated to 98°C for 10 min. Secreted extracellular protein sample was prepared by harvesting bacterial culture supernatant via centrifugation (5,000 $\times$ *g*, 5 min) and further clarified through a 0.2 $\mu$m filter. Protein was then precipitated using 20% (vol/vol) trichloroacetic acid at 4°C, washed by 100% ice-cold acetone, and dissolved in 100 $\mu$L SDS-PAGE sample buffer in either the presence or absence of DTT (or $\beta$-mercaptoethanol [$\beta$-ME]) and heated to 100°C for 5 min. Samples (5 to 10 $\mu$L) were then separated by electrophoresis with either a 12% (vol/vol) SDS-acrylamide gel or a 4% to 12% gradient gel (Thermofisher Scientific).

For the separation of the secreted extracellular protein contents in semidenatured and nondenatured conditions, extracellular protein samples were harvested and clarified as above, and proteins were precipitated with 60% (wt/vol final) saturated ammonium sulfate at 4°C overnight. The precipitated extracellular protein was collected via centrifugation (20,000 $\times$ *g*, 10 min, 4°C) and dissolved in PBS with

a 50-fold reduced volume. Sample as above (or affinity purified protein sample) were then mixed 1:1 with the SDS-PAGE sample buffer (semidenatured condition) or native PAGE sample buffer (62.5 mM Tris pH 6.8, 25% [vol/vol] glycerol, 1% [wt/vol] bromophenol blue) (nondenatured condition) in the presence or absence of 10 mM DTT, and unheated protein was separated by electrophoresis with a 4 to 12% Bis-Tris gel (Thermo Fisher Scientific) with MES running buffer (Thermo Fisher Scientific; semidenatured condition); or a hand-casted 8% (vol/vol) native acrylamide gel (315 mM Tris pH 8.5, 0.1% [wt/vol] ammonium persulfate [APS], 8% [vol/vol] acrylamide/bis), with native electrophoresis buffer (25 mM Tris pH 8.5, 192 mM glycine; nondenatured condition).

Western immunoblotting was done as previously described (11), proteins were detected by rabbit anti-IcsA polyclonal antibodies (pAbs, made in-house) (48), mouse anti-FLAG antibody (anti-DYKDDDDK antibody, Genscript), mouse anti-IpaD pAbs (gifted by Nikolai Petrovsky, Flinders University), or rabbit anti-SurA pAbs (gifted by Carol Gross, University of California) with Pierce ECL Western Blotting Substrate (Thermo Fisher Scientific).

**Chemical crosslinking.** For chemical crosslinking, affinity-purified IcsAp protein was mixed with 0.1 mM DSP (Sigma) in PBS and incubated at room temperature for 30 min. The reaction was quenched by 50 mM Tris (pH 7.0). Next, 10 mM DTT was added to reduce the protein cross-link when required. Samples were mixed with SDS-PAGE sample buffer without $\beta$-mercaptoethanol ($\beta$-ME) and subjected to SDS-PAGE.

**Cell infection assays.** All cell infection assays were described previously (11). Briefly, for cell adhesion assay, confluent HeLa cell monolayers were washed with Dulbecco's PBS (d-PBS) and 100 $\mu$L of *Shigella* bacteria grown at mid-exponential phase washed in MEM were spinoculated (500 $\times$ *g*, 5 min) onto the monolayer at a multiplicity of infection (MOI) of 100 and incubated for 15 min at 37°C. Unbound bacteria were removed by three PBS washes, and cell-associated bacteria were enumerated by resuspending infected monolayers in 0.1% (vol/vol) Triton X-100 and serial dilution plating onto agar plates.

For cell invasion assays, confluent HeLa cell monolayers were infected with *Shigella* bacteria as above and incubated for 45 min at 37°C. Monolayers were washed with d-PBS and extracellular bacteria were eliminated by incubating with MEM containing 50 $\mu$g/mL of gentamicin. The viable intracellular bacteria were enumerated above.

For plaque formation assays, MDCK-2 cells were seeded at low density (5 $\times$ 10$^4$ cells per 24-well) and cultured for 5 days to confluent monolayers. *S. flexneri* strains grown to mid-exponential phase were diluted 1:500 in Hank's balanced salt solution (HBSS) containing 0.1 M sodium citrate and a 250 $\mu$L bacterial suspension (10$^6$ cells/mL) was added per MDCK-2 monolayer pretreated with Hank's balanced salt solution (HBSS) containing 0.1 M sodium citrate (37°C, 1 h) and incubated for 1.5 h at 37°C with 5% CO$_2$ with gentle rotation every 15 min. At 1.5 h postinfection (pi), 3 mL of DMEM containing 0.5% SeaKem ME agarose (Lonza), 5% (vol/vol) FBS, and 50 $\mu$g/mL of gentamicin was added. At 48 h pi, a 3 mL second overlay containing the same medium as the first overlay supplemented with 0.1% (wt/vol) Neutral Red was added and incubated a further 2 h before imaging plaques. The plaque diameter was measured by Image J and plotted. For plaque formation with $\Delta$icsB strains, after 1.5 h pi, 3 mL of DMEM containing 5% (vol/vol) FBS, 100 $\mu$g/mL of gentamicin, and 50 $\mu$g/mL of kanamycin were added. The above medium was refreshed daily, and plaques were imaged and measured directly using a light microscope with a charge-coupled device (CCD) camera calibrated with a hemocytometer on day 2 postinfection (pi).

For intracellular F-actin staining, HeLa cells grown to $\sim$50% confluence on a coverslip were spinoculated (500 $\times$ *g*, 5 min) with 200 $\mu$L *Shigella* bacteria (10$^8$ cells) grown at mid-exponential phase (1:1000 diluted in MEM) and incubated for 1 h at 37°C with 5% CO$_2$. At 1 h pi, extracellular bacteria were inactivated by incubation with 500 $\mu$L MEM containing 50 $\mu$g/mL gentamicin for another 1 h. Coverslips were then washed three times with PBS and were used in subsequent experiments.

**Immunofluorescence microscopy.** For IcsA surface labeling, bacteria (10$^8$ cells) grown at mid-exponential phase were harvested via centrifugation (16,000 $\times$ *g*, 1 min), fixed with 3.7% (wt/vol) formaldehyde in PBS for 20 min at room temperature. Fixed bacteria were washed with PBS and a 5 $\mu$L suspension was then centrifuged onto coverslips precoated with 0.01% (wt/vol) poly-L-lysine (Sigma) in a 24-well tray (16,000 $\times$ *g*, 1 min). Coverslips were then incubated sequentially with rabbit anti-IcsA pAbs (1:100) or together with mouse anti-FLAG (1:100), then with anti-rabbit Alexa Fluor 488 antibody (Invitrogen, 1:100) or together with anti-mouse Alexa Fluor 647 (Invitrogen, 1:100) in PBS containing 10% (vol/vol) FBS (Gibco) with PBS washes in between. Coverslips were then mounted with ProLong Diamond Antifade Mountant (Invitrogen) and imaged using a ZEISS Axio Vert.A1 microscope.

For intracellular F-actin tail staining, corresponding samples described above were washed with d-PBS once and fixed with 3.7% (wt/vol) formaldehyde in PBS for 20 min at room temperature, quenched with 50 mM NH$_4$Cl in PBS for 10 min and permeabilized with 0.1% (vol/vol) Triton X-100 in PBS for 5 min. Samples were then blocked with 5% FBS in PBS for 20 min before being stained with Alexa Flour 594 phalloidin (1:100, Invitrogen) for 1 h and 10 $\mu$g/mL DAPI (Invitrogen) for 1 min. Samples were mounted and imaged as above.

**Proteinase accessibility assay.** Proteinase accessibility assay was described previously (6). Briefly, *Shigella* strains grown overnight were collected (1 $\times$ 10$^9$ cells) washed with and resuspended into 1 mL PBS and incubated with 33 nM human neutrophil elastase (hNE, Elastin Products) or proteinase K at 20 $\mu$g/mL (New England Biolabs) at 37°C. At indicated time points (see the figure legends), 100 $\mu$L samples were taken, mixed with an equal volume of SDS-PAGE sample buffer, and immediately incubated at 100°C for 15 min. Alternatively, digestion fractions at the indicated time point were centrifuged (16,000 $\times$ *g*, 1 min) to give whole bacterial and supernatant fractions, mixed with an equal volume of SDS-PAGE sample buffer, and immediately incubated at 100°C for 15 min. Samples were then subjected to SDS-PAGE and Western immunoblotting with anti-IcsA antibody or anti-FLAG antibody.

**Structure prediction and annotation.** IcsA structures were predicted using Colab Alpha-fold V2.1.0 structure prediction (28) built-in Chimera X1.3 (49) and annotated with Chimera X1.3.

**Data availability.** All data generated or analyzed during this study were included in this article and Supplemental File 1.

## SUPPLEMENTAL MATERIAL

Supplemental material is available online only.

**SUPPLEMENTAL FILE 1**, PDF file, 0.5 MB.

## ACKNOWLEDGMENTS

We thank Kenneth Beagley for providing mammalian cell lines for this study.

This study was supported by grants by the Australian National Health and Medical Research Council (NHMRC GNT1144046), the Clive and Vera Ramaciotti Foundations (2017HIG0119), and a Georgina Sweet Award for Women in Quantitative Biomedical Science to M.T. J.Q. received a Faculty of Science Postgraduate Scholarship from the University of Adelaide. This work made use of a microscopy facility supported by the Ian Porter Foundation. No funder had a role in the study design, data collection, analysis, decision to publish, or preparation of the manuscript.

J.Q. contributed to project conception and design, conducted experiments and contributed to data collection, analysis, and interpretation. M.T. supervised the study, contributed to data interpretation, and obtained funding. R.M. contributed to data interpretation and provided study reagents. J.Q. drafted the manuscript, and R.M., Y.H., and M.T. substantially revised the manuscript.

We declare no competing interests.

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
