## [Reviewer comments · Microbiology Spectrum]

Microbiology Spectrum

Cysteine dependent conformation heterogeneity of *Shigella flexneri* autotransporter IcsA and implications in its function

Jilong Qin, Yaoqin Hong, Renato Morona, and Makrina Totsika

Corresponding Author(s): Jilong Qin, Queensland University of Technology

Review Timeline:

Submission Date:	August 29, 2022
Editorial Decision:	September 30, 2022
Revision Received:	October 9, 2022
Accepted:	October 30, 2022

Editor: Swaine Chen

Reviewer(s): The reviewers have opted to remain anonymous.

Transaction Report:

DOI: <https://doi.org/10.1128/spectrum.03410-22>

September 30, 2022

Dr. Jilong Qin
Queensland University of Technology
Brisbane
Australia

Re: Spectrum03410-22 (Cysteine dependent conformation heterogeneity of *Shigella flexneri* autotransporter lcsA and implications in its function)

Dear Dr. Jilong Qin:

For Microbiology Spectrum, we are not making decisions based on potential impact or novelty - either for or against any paper. Thus, the comments from the transferred reviews that are related to impact of the work and any claims about either the novelty or the incremental nature of the results presented are not considered here.

There appear to be two main technical comments, one of which (loading controls) was mentioned by both previous reviewers. The manuscript claims that the mutants of lcsA have "no impact on lcsA production" (line 141, and reclaimed at line 345). This perhaps is of greater sensitivity because, as noted by the authors, cysteine mutations in other Type V autotransporter passenger domains do impact protein expression. For the first claim at line 141, referencing Figure 2e, I agree with the previous reviewers that some loading control will be required to justify this claim.

Furthermore, the previous reviewer 1 noted that cases of absence of detection of certain bands would warrant having a loading control; I find this to be a reasonable suggestion as well that is not sufficiently addressed by the arguments presented in the response to reviewers. Thus, the loading control request applies most clearly to Figure 2e, which is cited as the data supporting the claim of no impact on expression, but also applies to 2a and 2d. This issue of expression levels of course would impact the interpretation of one of the manuscript's main results in Figure 2b, and I believe that strengthening the expression data would in turn strengthen this result about differential antibody detection.

There do seem to be some cases where the result being claimed is a qualitative result that wouldn't depend on expression levels in bacteria. For these, I agree that explicit loading controls would not be necessary. These appear to be Figures 2c, 3a, 3d, all of Figure 4, and 6b. It remains unclear whether this would be strictly necessary for 3e (where the loading and/or expression of wt and C130S seems quite different and one may be expecting a dimer for the wt) and 6c, and I would leave it to the authors to either include it or justify why it is not necessary.

Regarding the second comment about showing the neutrophil elastase proteolysis of lcsA mutants in the delta-ipaD/delta-icsA strain - this data helps the paper, and in this case I also agree that loading controls are not necessary. While it seems that this experiment was not able to provide positive support for a T3SS-related conformational change in the cysteine mutants, previous authors (reference 6 by Brotcke-Zumsteg et al) were able to see such a change in a similar experiment with wt lcsA. It seems this was not seen in this manuscript and I suggest that a comment addressing this discrepancy be noted (and whether the authors would then make any further comments on the support or non-support of this experiment to their claims). i.e. this issue would likely be sufficient to address with just a revision to the text.

Two minor comments:

1. N/C-terminus is the noun, N/C-terminal is the adjective. The choice should be made to fit the usage throughout the manuscript.
2. The re-use of I and II in two different senses (different peaks in SEC and different bands in a western blot) is confusing in Figure 4 and I suggest some different symbols be used.

Link Not Available

Sincerely,

Swaine Chen

Journals Department
Reviewer comments:

Staff Comments:

Preparing Revision Guidelines

Please return the manuscript within 60 days; if you cannot complete the modification within this time period, please contact me. If you do not wish to modify the manuscript and prefer to submit it to another journal, please notify me of your decision immediately so that the manuscript may be formally withdrawn from consideration by Microbiology Spectrum.

Ref. Spectrum03410-22

06 Oct 2022

Re: "Cysteine dependent conformation heterogeneity of *Shigella flexneri* autotransporter IcsA and implications in its function"

Thank you for the opportunity to revise our manuscript for publication in *Microbiology Spectrum*.

We have taken the editor's comments on board and now provide additional data requested by previous reviewers with loading controls (new Figure 2a, 2d and 2e), and made modifications in text in our method section and figure legend accordingly. We also clarified that our hNE limited proteolysis assay does agree with previously published work by *Brotcke-Zumsteg et al* and provided evidence in our previous work with the same strains. These points were further clarified with modifications to the revised manuscript in track changes. We also provide a comprehensive response to editor explaining why loading controls in our Native PAGE experiment (Figure 3e and 6c) do not influence our analysis. We have also identified a mislabelling in our Figure 5b and now supplied with a revised Figure 5b.

We believe that the changes incorporated as part of the review process have

improved our manuscript and hope that you will now find it suitable for publication.

A point-by-point response to editor comments is provided together with a version of the revised manuscript including tracked changes.

We look forward to hearing back from you soon.

Yours sincerely,

Dr Jilong Qin (corresponding author)

Dr Makrina Totsika (corresponding author)

For Microbiology Spectrum, we are not making decisions based on potential impact or novelty - either for or against any paper. Thus, the comments from the transferred reviews that are related to impact of the work and any claims about either the novelty or the incremental nature of the results presented are not considered here.

There appear to be two main technical comments, one of which (loading controls) was mentioned by both previous reviewers. The manuscript claims that the mutants of IcsA have "no impact on IcsA production" (line 141, and reclaimed at line 345). This perhaps is of greater sensitivity because, as noted by the authors, cysteine mutations in other Type V autotransporter passenger domains do impact protein expression. For the first claim at line 141, referencing Figure 2e, I agree with the previous reviewers that some loading control will be required to justify this claim.

Response:

We thank the Editor and previous reviewers for their constructive comments. As requested, we have now repeated protein expression analyses described in Figure 2e to include loading controls by immunoblotting with anti-SurA (revised Figure 2e). Results confirmed our original claim that cysteine substitution in IcsA does not affect IcsA expression level.

Furthermore, the previous reviewer 1 noted that cases of absence of detection of certain bands would warrant having a loading control; I find this to be a reasonable suggestion as well that is not sufficiently addressed by the arguments presented in the response to reviewers. Thus, the loading control request applies most clearly to Figure 2e, which is cited as the data supporting the claim of no impact on expression, but also applies to 2a and 2d. This issue of expression levels of course

would impact the interpretation of one of the manuscript's main results in Figure 2b, and I believe that strengthening the expression data would in turn strengthen this result about differential antibody detection.

Response:

As requested, we have now repeated blots shown in Figure 2a and 2d to include appropriate loading controls, i.e. immunoblotting of whole cell lysates with anti-SurA (revised Figure 2a), and of precipitated extracellular proteins with anti-IpaD (revised Figure 2d). We have modified text in our manuscript to closely reflect our data as: " Indeed, when we precipitated the secreted extracellular protein from culture supernatants, we found all IcsA cysteine substitution mutants were able to secrete cleaved IcsAp (Figure 2d), confirming that passenger translocation was not affected. Results confirmed our original claim that FLAG tag insertion in IcsA does not affect IcsA biogenesis (Figure 2a) and that cysteine substitution does not lead to an IcsA translocation defect (Figure 2d). For figure 2e, please see our response above. As such our interpretation of a differential antibody detection remains well supported.

There do seem to be some cases where the result being claimed is a qualitative result that wouldn't depend on expression levels in bacteria. For these, I agree that explicit loading controls would not be necessary. These appear to be Figures 2c, 3a,

3d, all of Figure 4, and 6b. It remains unclear whether this would be strictly necessary for 3e (where the loading and/or expression of wt and C130S seems quite different and one may be expecting a dimer for the wt) and 6c, and I would leave it to the authors to either include it or justify why it is not necessary.

Response:

We thank the Editor for their detailed assessment of our expression analyses. While we agree that loading controls were appropriate for denatured protein analyses shown in Fig 2a, d and e (now revised), we argue that they would not be appropriate for native PAGE analyses presented in Figures 3e and 6c. Secretion levels of IcsAp mutants could be accurately compared and confirmed to be the same in Figure 2d as a method that ensures robust harvesting efficiency for extracellular proteins could be used for denatured proteins (acid (TCA) precipitation). In contrast, the method used to harvest secreted IcsAp from culture supernatant in their native conformation(s), as shown in Figures 3e and 6c, was by precipitating protein using 60% saturated ammonium sulfate. The efficiency of protein precipitation via ammonium sulfate is influenced by exposure of protein surfaces, therefore IcsA proteins present in different conformations will be harvested in different yields. This technical matter however, does not affect our interpretation and analysis of differing IcsAp conformational landscapes, as we are not comparing the quantity of each specific conformation population across samples, rather, we are comparing the major conformation population adopted by

each different IcsAp mutant and the migration patterns between samples precipitated under the same condition. (i.e. for IcsA^{WT}, and IcsA^{C130S} the major conformational population is IcsAp-ox; while for the paired cys IcsA mutants, the major bands become dimer, IcsA-re, and an additional C375S-ox for IcsA^{C375S}, Figure 3e). Thus, potential loading differences should not influence our data analysis and interpretation. Our data (Figure 3e and 6c) hence clearly show that substitution of C375 and/or C379, but not C130 alters secreted IcsAp migration patterns in comparison to WT, regardless of each band's intensity, supporting our conclusion that the central pair of cysteines is responsible for determining the IcsA conformational landscape. We believe that the dimer population for IcsA^{WT} is present in relatively low abundance (Figure 3d) and can thus not effectively be detected by native-PAGE in Figures 3e and 6c. This is potentially due to less favorable dimer formation between the intact pair cysteine residues in IcsA^{WT} and IcsA^{C130S}, which control the IcsA conformational landscape.

Regarding the second comment about showing the neutrophil elastase proteolysis of IcsA mutants in the delta-ipaD/delta-icsA strain - this data helps the paper, and in this case I also agree that loading controls are not necessary. While it seems that this experiment was not able to provide positive support for a T3SS-related conformational change in the cysteine mutants, previous authors (reference 6 by Brotcke-Zumsteg et al) were able to see such a change in a similar experiment with

wt lcsA. It seems this was not seen in this manuscript and I suggest that a comment addressing this discrepancy be noted (and whether the authors would then make any further comments on the support or non-support of this experiment to their claims). i.e. this issue would likely be sufficient to address with just a revision to the text.

Response:

We appreciate the point raised by the Editor but would like to point out that our results do not in fact disagree with previous work by Brotcke-Zumsteg et al (2014). In order to accurately detect protein conformational differences by limited proteolysis, samples need to be digested in the same way within the same experiment (i.e. immunoblotted in the same way and exposed together). We used this approach in Figure 3a and Figure S2 to compare each lcsA mutant directly to the WT and all samples were processed and imaged in the same way. To observe lcsA conformational differences between the *ΔipaD* and the WT background, lcsA expressed in each of these strain backgrounds needs to be treated within the same experiment and blotted and exposed together. We have previously confirmed this to be the case with these strains and lcsA construct (Figure 2, Chapter 4, Page 121, [https://digital.library.adelaide.edu.au/dspace/bitstream/2440/123929/1/Qin2020_PhD.pdf]), where we showed that a 40 kD Flag-tagged lcsA fragment expressed in the *ΔipaD* strain has increased resistance to hNE. We have now added a sentence to clarify this in the manuscript text as follows: "We have also confirmed the lcsA

conformational changes when expressed in our *S. flexneri* $\Delta ipaD\Delta icsA$ background via limited proteolysis (33)". In this study we are focusing on conformational differences between IcsA WT and IcsA mutants expressed in the same strain background. As such, our experiments were not designed or optimised to detect differences between *S. flexneri* $\Delta icsA$ and *S. flexneri* $\Delta ipaD\Delta icsA$ and comparisons of these nature to previously reported data are not deemed valid.

Two minor comments:

1. N/C-terminus is the noun, N/C-terminal is the adjective. The choice should be made to fit the usage throughout the manuscript.

Response:

We have now corrected the use of N/C-terminus and N/C-terminal throughout our manuscript.

2. The re-use of I and II in two different senses (different peaks in SEC and different bands in a western blot) is confusing in Figure 4 and I suggest some different symbols be used.

Response:

We would like to clarify that the choice to label peaks and bands with identical symbols was intentional as the purified IcsA protein samples (peaks) in SEC (labelled as fraction I and II in Figure 4b) were the same samples used in the Coomassie stained PAGE gels shown in Figure 4c and 4d. We believe keeping the labelling consistent between the panels is important in helping readers interpret our data clearly and we have thus kept the labelling unchanged. To avoid confusion we have now clarified this in the Figure 4 legend.

October 18, 2022

Dr. Jilong Qin
Queensland University of Technology
Brisbane
Australia

Re: Spectrum03410-22R1 (Cysteine dependent conformation heterogeneity of *Shigella flexneri* autotransporter lcsA and implications in its function)

Dear Dr. Jilong Qin:

Your manuscript has been accepted, and I am forwarding it to the ASM Journals Department for publication. You will be notified when your proofs are ready to be viewed.

Sincerely,

Swaine Chen
Editor, Microbiology Spectrum
